# OpenReview forum: "SG2Loc: Sequential Visual Localization on 3D Scene Graphs"
_ICLR.cc/2026/Conference — ICLR 2026 Conference Withdrawn Submission_

### Official Review · Reviewer_2rVs · 2025-10-19

**Soundness:** 2
**Presentation:** 3
**Contribution:** 1
**Rating:** 2
**Confidence:** 5

**Summary:**

This paper proposes a method for sequential visual localization based on compact 3D scene graphs. The approach models environments as graphs, where nodes represent coarse object meshes and edges encode spatial relations. During localization, semantic features extracted from image patches are compared with projected object identities within a particle filter framework to estimate camera poses. The method is evaluated on real-world datasets and reports comparable localization accuracy while reducing storage requirements.

**Strengths:**

The paper is clearly structured and generally easy to follow. The idea of leveraging a compact scene graph for sequential localization is interesting, and the paper demonstrates careful engineering and integration of existing components.

**Weaknesses:**

1. Motivation and scope.

The motivation of the paper is not clearly aligned with the proposed approach. The authors argue that existing scene coordinate regression (SCR) and absolute pose regression (APR) methods struggle in complex, large-scale environments (line 91, page 2), yet the experiments are limited to small, static indoor datasets such as ScanNet and 3RScan, which do not convincingly demonstrate the claimed advantages. In contrast, prior works—including SCR and APR methods—have been evaluated on more challenging benchmarks such as Cambridge Landmarks and Aachen-Day-Night, which involve illumination changes, dynamic scenes, and weather variations. In addition, according to Table 3, the proposed approach does not show a clear advantage over SCR methods in terms of storage. It is also worth noting that many existing visual localization approaches operate without explicit gravity alignment or auxiliary sensors, so the benefits of the proposed method are not fully clear.

2. Missing Related Work and Limited Novelty.

The technical novelty of this paper over SceneGraphLoc appears limited, as the proposed framework mainly extends it to handle image sequences. Furthermore, the paper omits discussion and comparison with several relevant recent works. The statement (line 92, page 2) that the method avoids storing image databases or point clouds is not a significant advantage, since many modern methods—such as SCR, APR approaches—already share this property. Moreover, these existing methods typically support full 6-DoF pose estimation, whereas the proposed method appears to handle only 4-DoF localization.

In addition, comparisons with more recent and representative baselines are missing. For instance, GLACE [1] and R-SCoRe [2] have demonstrated strong results on large-scale benchmarks (e.g., Cambridge, Aachen, Hyundai Department Store), and differentiable representation-based GS-CPR [3] achieves efficient pose refinement without image databases and point clouds. Including such baselines would make the experimental evaluation more convincing and better clarify the contribution of this work.

[1] GLACE: Global Local Accelerated Coordinate Encoding, CVPR 2024

[2] R-SCoRe: Revisiting Scene Coordinate Regression for Robust Large-Scale Visual Localization, CVPR 2025

[3] GS-CPR: Efficient Camera Pose Refinement via 3D Gaussian Splatting, ICLR 2025

**Questions:**

If the authors believe that existing SCR or APR methods perform poorly on large and complex scenes, I strongly encourage including at least two larger and more challenging benchmarks—such as Cambridge Landmarks, Hyundai Department Store, or Aachen-Day-Night—and comparing against GS-CPR, GLACE, and R-SCoRe.

If the key contribution is improved storage efficiency or the avoidance of image/point cloud databases, please provide more thorough comparisons with recent differentiable-representation-based methods and discuss these aspects more explicitly in the related work and experiments.

Why does this paper not provide the recommended Reproducibility statement and the Use of Large Language Models (LLMs) statement?

---

> ### Author Response · Authors · 2025-11-28
> **Rebuttal**
>
> **W1:**
> - **The motivation (limitations of SCR and APR in large, complex scenes) does not match the evaluation, which uses only small indoor datasets (ScanNet, 3RScan). Prior SCR and APR works are tested on more challenging benchmarks (Cambridge, Aachen).**
> We thank the reviewer for this remarks. The referenced benchmarks (Cambridge Landmarks, Aachen Day-Night) are outdoor datasets, where sequential localization is largely solved and GPS is typically available. Our focus is on indoor localization, which is substantially more challenging due to clutter, occlusions, frequent scene changes, and the absence of GPS. We acknowledge that this was not stated clearly enough and have revised the abstract and introduction accordingly in lines 60-61.
>
>
> - **W1 and W2: The storage advantage over SCR methods is unclear from Table 3.**
> Regarding storage, SCR methods such as ACE are memory-efficient. However, our results on 3RScan, a dataset with significant temporal change, motion blur, and ambiguous viewpoints, show that SG2Loc achieves higher accuracy than ACE (Tables 1-2) on that dataset.
>
> - **W1 and W2: Existing methods typically support full 6-DoF pose estimation, whereas the proposed method appears to handle only 4-DoF localization.**
> In sequential localization, the gravity direction is almost always available from mobile devices (e.g., smartphones, mobile robots, AR/VR headsets), making 4-DoF a practical and commonly used setting. Our goal is to exploit this information to improve robustness and efficiency. Moreover, HLoc and ACE do not provide a straightforward way to integrate a known gravity constraint.
>
> **W2: The technical novelty of this paper over SceneGraphLoc appears limited, as the proposed framework mainly extends it to handle image sequences.**
> We thank the reviewer for the comment. To clarify, our method does not extend SceneGraphLoc. SceneGraphLoc performs coarse scene retrieval, whereas our work tackles a different problem, that is estimating the 4-DoF camera pose using sequential probabilistic reasoning. Our particle-filter formulation integrates semantic, photometric, and geometric cues specifically for fine-grained pose estimation, not scene-level retrieval. Only the minor contribution in chapter 4 extends SceneGraphLoc.
>
> **Q1: If the authors believe that existing SCR or APR methods perform poorly on large and complex scenes, I strongly encourage including at least two larger and more challenging benchmarks—such as Cambridge Landmarks, Hyundai Department Store, or Aachen-Day-Night—and comparing against GS-CPR, GLACE, and R-SCoRe.**
> We thank the reviewer for raising this point. Aachen provides no sequences and both Aachen and Cambridge are outdoor datasets, making them unsuitable for our sequential indoor setting. We compare to 3RScan, which includes temporal scene changes, and ScanNet, a standard indoor dataset, to evaluate robustness in the intended use case.
>
> **Q2: Why does this paper not provide the recommended Reproducibility statement and the Use of Large Language Models (LLMs) statement?**
> We thank the reviewer for pointing that out, we have added this section (chapter 7). We have used LLM for polishing text (grammar, wording). Regarding reproducibility, we will publish the code, such that the results can be reproduced.

---

> > ### Author Response · Authors · 2025-12-03
> > **Rebuttal**
> >
> > **Additional baseline GS-CPR**
> >
> > We thank the reviewer for suggesting additional baselines. We implemented the GS-CPR (Liu et al., ICLR 2025) refinement for ACE poses. The results on the 3RScan dataset for all sequence lengths (5/10/25/50) are shown in the table below. **ACE + GS-CPR** improves ACE, but our method still achieves better accuracy than both baselines (↓ = lower is better, ↑ = higher is better). We have added those results to the main tables in the paper.
> >
> > | Seq. Length | Method           | Median Pos m (↓)  | Median Rot ° (↓) | Recall @0.25m, 2° (↑) |
> > |-----------------|------------------|--------------|--------------|----------------|
> > | 5           | ACE              | 1.66            | 42.38          | 0.11                    |
> > | 5           | ACE + GS-CPR     | 1.61            | 50.13          | 0.14
> > | 5          | **SG2Loc (Ours)** | **0.20**         | **3.27**        | **0.32**
> >
> > | Seq. Length | Method           | Median Pos m (↓) | Median Rot ° (↓) | Recall @0.25m, 2° (↑)|
> > |-----------------|------------------|--------------|--------------|----------------|
> > | 10           | ACE              | 0.93 | 27.04| 0.14                    |
> > | 10           | ACE + GS-CPR     | 0.75 | 20.88 | 0.20                   |
> > | 10          | **SG2Loc (Ours)** | **0.18** | **2.96** | **0.33**                |
> >
> > | Seq. Length | Method           | Median Pos m (↓) | Median Rot ° (↓) | Recall @0.25m, 2° (↑)|
> > |-----------------|------------------|--------------|--------------|----------------|
> > | 25           | ACE              | 0.39 | 10.31 | 0.21
> > | 25           | ACE + GS-CPR     | 0.18 | 5.93 | 0.27 |
> > | 25          | **SG2Loc (Ours)** | **0.14** | **2.73** | **0.39**
> >
> > | Seq. Length | Method           | Median Pos m (↓) | Median Rot ° (↓) | Recall @0.25m, 2° (↑)|
> > |-----------------|------------------|--------------|--------------|----------------|
> > | 50           | ACE              | 0.19 | 5.33 | 0.29
> > | 50           | ACE + GS-CPR     | 0.14 | 4.37 | 0.36
> > | 50          | **SG2Loc (Ours)** | **0.10** | **2.62** | **0.43**

---

### Official Review · Reviewer_sH4f · 2025-10-30

**Soundness:** 3
**Presentation:** 3
**Contribution:** 2
**Rating:** 2
**Confidence:** 4

**Summary:**

This paper presents a lightweight visual localization method for sequential image inputs, utilizing compact 3D scene graphs as the scene representation.
In a 3D scene graph, each node corresponds to an object instance, with each instance characterized by a set of multi-modal attributes including RGB frames, a point cloud, textual annotation and a coarse 3D mesh.

The authors formulate the camera localization as a particle filtering problem, where particle weights are determined by combining semantic similarity, the depth and color scores between the query image and the map projection.
The proposed approach allows for pose searching and optimization over multiple iterations, with further refinement achieved through *RANSAC-PnP*.

Experimental results on two public datasets demonstrate that the method achieves competitive localization performance while significantly reducing storage requirements.

**Strengths:**

- The concept of leveraging 3D semantic scene graphs for camera relocalization is innovative and well-justified. By compactly integrating high-level semantic relationships with geometric representations, they provide a sufficiently informative foundation for accurate localization.
- The task is formulated as an iterative particle filtering problem, where particle weights are computed through multi-modal comparisons between the query and the rendered representations (semantic, depth, and color maps).

**Weaknesses:**

1. The core system is conceptually simple: a standard particle filter in $SE(3)$ with a similarity-based observation model. Particle filtering for robot localization has been extensively studied, including with visual or learned models.  Although simplicity is not a drawback in itself, I'm questioning the suitability of particle filtering for this specific task:
	1. The framework relies on a considerable number of hyper-parameters, especially the initial particle sampling strategy, which uses four predefined heights, appears quite contrived to me.
	2. The multi-round particle optimization and ray tracing operations are computationally expensive, potentially limiting the method's adoption in real-time or resource-constrained applications
2. The evaluation may be not entirely fair and convincing. While the method is designed for sequential input, it is only compared against baselines targeting per-frame localization. Besides, the direct use of ground-truth poses provided by the datasets as ego-motion raises another major concern on its practical performance.
3. It is suggested to provide more details on how these baselines are adapted and implemented, as the statistics reported are questionable - particularly in `Tab.1`, for *HLoc* and *MeshLoc* that rely on image retrieval, the average position errors seem unexpectedly large.
4. The claimed low-memory benefit and performance gain are not sufficiently compelling. Established scene coordinate regression methods, like *ACE*, *GLACE* and *R-SCoRe*, demonstrate similar or even better savings while achieving  top performance within minutes—and crucially, *without relying on additional priors such as depth, meshes, or annotations*.

**Questions:**

1. I remain concerned about the unexpectedly large average pose errors reported for *HLoc* and *MeshLoc*. The authors should verify their implementation. It is also recommended to constrain the final pose estimates within the scene's bounding box as a straightforward sanity check.
2. The selection criteria for a fix-length of sequential inputs remains unclear to me. For instance, it is not specified whether frame downsampling is applied to form the query sequence, or how frames in a sequence are chosen. This pre-processing step should be clearly detailed.
3. The results in `Tab.8` indicate that semantic signals are the most contributive factor, which is not fully straightforward to me, since intuitively depth cues are more important.

---

> ### Author Response · Authors · 2025-11-28
> **Rebuttal**
>
> **W1. The system is conceptually simple. It depends on several hyperparameters, including fixed heights (1.1) multi-round optimization and raycasting make it computationally expensive (1.2).**
> Hyperparameters: All parameters, including the four height levels used for initialization, are fixed across all experiments. With these fixed settings, performance improves. The chosen heights are typical for indoor datasets. We have highlighted this in the paper (lines 283-284).
>
> Runtime and real-time use: While multi-round optimization and raycasting are computationally heavier than retrieval, the keyframing experiment (Table 6) shows that SG2Loc runs in real time when updating at a reduced frame rate, without loss in accuracy. This makes the approach suitable for resource-constrained settings.
>
> **W2:**
> **... While the method is designed for sequential input, it is only compared against baselines targeting per-frame localization.**
> We thank the reviewer for the comment. We include Loc-NeRF, a sequential baseline, to provide a direct comparison in a similar setting (Table 1 and 2). For the per-frame baselines, we follow the standard implementation (HLoc Sarlin et al. CVPR 2019, MeshLoc Panek et al. ECCV 2022) and make them “sequential” by selecting, for each sequence, the frame with the highest RANSAC inlier count, ensuring a fair comparison.
>
> **Besides, the direct use of ground-truth poses provided by the datasets as ego-motion raises another major concern on its practical performance.**
> We thank the reviewer for these questions. The ego-motion between frames can be obtained from any SLAM system. In our experiments, Scan3R provides poses from Google Tango, while ScanNet uses the Structure Sensor for RGB-D scanning. We compute relative motion from these poses and model their uncertainty by adding Gaussian noise (0.05 m, 0.05 rad), as noted in lines 216–217. All baselines use this quasi-ground truth for backpropagating the highest-inlier pose.
>
> We additionally report results using _DROID-SLAM_ (Teed & Deng, 2021, NeurIPS) for the relative motion on _ScanNet_ (sequence length 5) instead of the quasi ground truth. The performance is comparable, showing that SG2Loc is robust to moderate drift in the motion estimate:
>
>
> | Method                    | Median Pos (m) | Median Rot (°) | Recall @ 0.25m, 2° |
> |---------------------------|----------------|----------------|-------------------------|
> | HLoc                      | **0.09**       | 2.63           | **0.38**                |
> | HLoc w/ SLAM poses        | 0.15           | 2.67           | 0.34                    |
> | **SG2Loc (Ours)**         | 0.12           | **2.55**       | 0.36                    |
> | **SG2Loc w/ SLAM poses**  | 0.16           | 2.63           | 0.36                    |
>
> We have added that experiment to the supplement chapter F.
>
> **W3: ... the statistics reported are questionable - particularly in Tab.1, for HLoc and MeshLoc that rely on image retrieval, the average position errors seem unexpectedly large.**
> We thank the reviewer for the concern. We use the default implementations of HLoc (Sarlin et al. CVPR 2019) and MeshLoc (Panek et al. ECCV 2022). The only modification is that 2D-3D correspondences are obtained via raycasting the mesh to make the comparison fair. Neither HLoc nor MeshLoc can bound the space. Standard localization metrics are recall and median errors, we additionally present the average to illustrate these failure cases.
>
> **W4: The claimed low-memory benefit and performance gain are not sufficiently compelling... ACE, GLACE and R-SCoRe achieve top performace fast, with low memory and without relying on depth, meshes, or annotations.**
> We include ACE as a representative scene-coordinate regression baseline. While ACE is indeed memory-efficient, its performance on 3RScan, a dataset with significant temporal variation and object rearrangements, is noticeably lower (Tables 1-2).
>
> **Q2: The selection criteria for a fix-length of sequential inputs remains unclear to me. For instance, unclear how frames in a sequence are chosen.**
> We thank the reviewer for pointing that out. The fixed sequence lengths (5, 10, 25, 50) are used to illustrate the effect of longer temporal context, SG2Loc can operate on sequences of arbitrary length. As described in lines 348-356, we localize every frame on 3RScan and every 25th frame on ScanNet (following SceneGraphLoc).
>
> **Q3: The results in Tab.8 indicate that semantic signals are the most contributive factor, which is not fully straightforward to me, since intuitively depth cues are more important.**
> We thank the reviewer for that observation. While depth is helpful, the filter first needs a coarse pose over a large search space, where semantic identities give strong global cues. For example, recognizing a chair or nightstand can be more discriminative than depth alone. After this stage, we refine the pose with dense matching and RANSAC PnP.

---

### Official Review · Reviewer_dsUg · 2025-11-01

**Soundness:** 3
**Presentation:** 3
**Contribution:** 2
**Rating:** 4
**Confidence:** 3

**Summary:**

This paper introduces SG2Loc, a method for sequential visual localization that relies on 3D scene graphs instead of traditional dense 3D maps or large image databases.

The key idea is to represent an environment as a compact graph of object nodes and spatial relationships, where each node includes a coarse mesh and a semantic embedding.

Localization is formulated as a particle filtering problem that refines the camera pose over time.
The method achieves comparable accuracy to classical approaches such as HLoc and MeshLoc while requiring 10× less storage.

**Strengths:**

1. Overall, the paper is well-written and clearly structured.

2. The work extends the use of scene graphs in visual localization to the sequential setting, which is a meaningful and natural progression for this line of research.

3. While the use of a particle filter is not new, the paper integrates it effectively with semantic and geometric cues from scene graphs, resulting in an effective approach.

4. The method is compared against well-known baselines and achieves comparable performance while requiring less storage (with the exception of ACE).

5. The authors include thorough ablation studies that help clarify the impact of key design choices.

**Weaknesses:**

1. The contribution is primarily an integration of existing components, scene graphs, particle filtering, and standard similarity measures, rather than a fundamentally novel idea.

2. The method relies on pre-built, labeled 3D scene graphs with coarse object meshes, but the paper does not discuss how such graphs are generated.

3. The system estimates only 4 degrees of freedom (assuming known gravity direction), making it unsuitable for general 6-DoF localization and resulting in an unfair comparison with fully 6-DoF baselines such as HLoc and ACE.

4. The approach is considerably slower than retrieval-based methods.

5. Experiments are conducted only on indoor datasets, leaving their performance in outdoor scenes unexplored.

6. While the paper emphasizes low storage requirements, it does not discuss or compare to other memory-efficient visual localization methods such as SceneSqueezer [A] and [B].

-[A] Scenesqueezer: Learning to compress scene for camera relocalization. CVPR 2022.

-[B] Differentiable product quantization for memory efficient camera relocalization. ECCV 2024.

**Questions:**

1. Regarding W3, can you please show an experiment where you estimate the gravity direction using GeoCalib (Veicht et al., 2024)?

2. Please clarify how you construct the scene graph for each scene.

3. Please discuss/compare the memory-efficient methods mentioned in W6.

4. Is it possible to show a small experiment on an outdoor scene?

minor: In line 199, there is a small mistake: a 14×14 grid results in 196 patches, not 144 as mentioned.

---

> ### Author Response · Authors · 2025-11-28
> **Rebuttal**
>
> **W1: The contribution is primarily an integration of existing components, scene graphs, particle filtering, and standard similarity measures, rather than a fundamentally novel idea.**
> We respectfully disagree. While the system builds on established components, the core idea of using semantic object embeddings from a scene graph as the primary signal for fine localization is, to our knowledge, new. Prior work with scene graphs focuses on coarse retrieval. Here, semantics are used directly within a sequential particle-filter framework for pose estimation, which has not been explored before.
>
> **W2 and Q2: Please clarify how you construct the scene graph for each scene.**
> For 3RScan, annotated scene graphs are already provided with the dataset. For ScanNet, we follow SceneGraphLoc and generate scene graphs using SceneGraphFusion (Line 348-356 in the paper).
>
> **W3: The system estimates only 4 degrees of freedom (assuming known gravity direction), making it unsuitable for general 6-DoF localization and resulting in an unfair comparison with fully 6-DoF baselines such as HLoc and ACE.**
> In sequential localization, the gravity direction is almost always available from standard mobile devices (e.g., smartphones, mobile robots, AR/VR headsets), making 4-DoF a practical and commonly used setting. Our goal is to exploit this information to improve robustness and efficiency. Moreover, HLoc and ACE do not provide a straightforward way to integrate a known gravity constraint.
>
> **W4: The approach is considerably slower than retrieval-based methods.**
> Per-frame integration is slower, but SG2Loc is not dependent on frame-to-frame overlap. This allows us to process only selected frames and match the localization rate to our runtime budget. As shown in our keyframing experiment, the method remains *real-time in practice* when updating at a reduced frame rate, while maintaining better accuracy than baselines.
>
> **W5 and Q4: Experiments are conducted only on indoor datasets, leaving their performance in outdoor scenes unexplored.**
> We thank the reviewer for the comment. Our method is designed for indoor localization, and we acknowledge that this was not communicated clearly enough in the paper. Scene graphs with dense semantic annotations are currently available primarily for indoor environments, where localization is also more challenging due to clutter, occlusion, and the absence of GPS. Outdoor settings typically rely on GPS and large-scale maps rather than object-level scene graphs. We have made this scope clearer in the abstract and the main paper lines 60-61.
>
> **W6 and Q3: While the paper emphasizes low storage requirements, it does not discuss or compare to other memory-efficient visual localization methods such as SceneSqueezer [A] and [B].**
> We thank the reviewer for these interesting pointers and have added the suggested works to the related-work section. We also note that this is why ACE is included as a baseline, because it is a low-memory method.
>
> **minor: In line 199, there is a small mistake: a 14×14 grid results in 196 patches, not 144 as mentioned.**
> We thank the reviewer, we have corrected that.

---

### Official Review · Reviewer_M57c · 2025-11-05

**Soundness:** 2
**Presentation:** 2
**Contribution:** 2
**Rating:** 4
**Confidence:** 4

**Summary:**

The paper introduces SG2Loc, a sequential visual localization framework that utilizes 3D scene graphs as the underlying map representation. Unlike prior image- or point-cloud-based approaches, SG2Loc employs a particle filter operating on a 4-DoF state space, where each particle’s observation likelihood is computed by comparing ray-casted object predictions against semantic patch embeddings derived from a SceneGraphLoc-style encoder. The method further integrates SSIM and depth consistency terms, adopts coarse-to-fine search and KLD-based adaptive resampling, and refines the final pose using RoMa + PnP. The main claim is that SG2Loc achieves competitive localization accuracy with a dramatically smaller storage footprint, demonstrating the viability of semantic scene-graph maps for sequential localization tasks.

**Strengths:**

S1. The paper makes a clear and logical extension of SceneGraphLoc from single-frame reasoning to sequential probabilistic localization. The integration of semantic, photometric, and geometric cues in a unified particle-filter framework is well motivated and technically sound.

S2. The overall system is well-structured, including motion prediction, adaptive resampling, and pose refinement. The coarse-to-fine search is a sensible addition that improves robustness in practice.

S3. Experimental results indicate that the method achieves accuracy comparable to strong baselines while using significantly less map storage, supporting its motivation for efficient localization.

**Weaknesses:**

W1: The system mainly combines existing components, such as scene-graph-based embeddings,  particle filtering, SSIM/depth fusion, and RoMa+PnP refinement, without introducing a fundamentally new algorithmic contribution. The conceptual leap beyond SceneGraphLoc is relatively small.

W2: The semantic likelihood assigns a high score only when the predicted object ID from ray-casting matches the predicted object ID from the image patch. This hard matching assumption ignores soft uncertainty in detection and segmentation, which could make the system brittle to misclassification, occlusion, or open-set objects. A more principled probabilistic treatment would have been preferable.

W3: The paper combines semantic, photometric, and depth likelihoods, but does not specify how these are normalized or weighted. Without clear scale calibration or hyperparameter justification, the combined likelihood may behave unpredictably across scenes.

W4. The transition model seems to rely on ego-motion estimation between frames, but details on how this motion is obtained are unclear.

**Questions:**

Q1. How robust is SG2Loc to semantic segmentation errors or to dynamic scenes where object layouts change?

Q2. How is ego-motion estimated between frames for the particle filter’s prediction step?

Q3. How are the weights between semantic, photometric, and depth likelihoods tuned? Are they fixed, or learned?

Q4. How sensitive is the performance to sequence length and particle count?

---

> ### Author Response · Authors · 2025-11-28
> **Rebuttal**
>
> **S1: The paper makes a clear and logical extension of SceneGraphLoc from single-frame reasoning to sequential probabilistic localization.**
> We thank the reviewer for the comment. Only the minor contribution in Chapter 4 extends SceneGraphLoc, the main contribution does not. SceneGraphLoc targets coarse scene retrieval, whereas our work estimates a 4-DoF camera pose via sequential probabilistic reasoning. Our particle filter fuses semantic, photometric, and geometric cues for fine-grained pose estimation, not scene-level retrieval.
>
> **S3: Experimental results indicate that the method achieves accuracy comparable to strong baselines while using significantly less map storage...**
> We thank the reviewer for this observation. In addition to the storage benefits, we would like to highlight that SG2Loc achieves strong accuracy even under keyframing, where localization is performed only on selected frames in real-time. As shown in our keyframing experiment (Table 6 main paper), SG2Loc outperforms HLoc in this setting while maintaining real-time performance and low storage.
>
> **W1: The system mainly combines existing components, ... , without introducing a fundamentally new algorithmic contribution. The conceptual leap beyond SceneGraphLoc is relatively small.**
> We would like to clarify that our method tackles a different problem from SceneGraphLoc. SceneGraphLoc performs coarse scene retrieval, whereas we address 4-DoF camera pose estimation directly on a 3D scene graph. Using semantic object identities as the observation signal for pose estimation and integrating them within a sequential particle-filter pipeline is, to our knowledge, *new*.
>
> **W2: The semantic likelihood relies on hard object-ID matches between ray-casting and image patches, ignoring uncertainty in detection or segmentation. This makes the system brittle to misclassification, occlusion, and open-set objects.**
> We thank the reviewer for the comment. This appears to be a misunderstanding: we do not use object IDs, but learned object embeddings. We check for alignment between predicted identities, but these identities are obtained via cosine similarity between patch embeddings and scene-graph object embeddings. Since the semantic categories are adapted to the target map, open-set ambiguity is reduced. Patches are compared only against objects that actually exist in the scene graph.
>
> **W3 and Q3: How are the weights between semantic, photometric, and depth likelihoods tuned? Are they fixed, or learned?**
> All losses are weighted equally, the weights are fixed and all parameters are kept fixed across all experiments. We have made this clearer in the paper in lines 283-284.
>
> **W4 and Q2: How is ego-motion estimated between frames for the particle filter’s prediction step?**
> We thank the reviewer for these questions. The ego-motion between frames can be obtained from any SLAM system. In our experiments, Scan3R provides poses from Google Tango, while ScanNet uses the Structure Sensor for RGB-D scanning. We compute relative motion from these poses and model their uncertainty by adding Gaussian noise (0.05 m, 0.05 rad), as noted in lines 216–217. All baselines use this quasi-ground truth for backpropagating the highest-inlier pose.
>
> We additionally report results using _DROID-SLAM_ (Teed & Deng, 2021, NeurIPS) for the relative motion on _ScanNet_ (sequence length 5) instead of the quasi ground truth. The performance is comparable, showing that SG2Loc is robust to moderate drift in the motion estimate:
>
>
> | Method                    | Median Pos (m) | Median Rot (°) | Recall @ 0.25m, 2° |
> |---------------------------|----------------|----------------|-------------------------|
> | HLoc                      | **0.09**       | 2.63           | **0.38**                |
> | HLoc w/ SLAM poses        | 0.15           | 2.67           | 0.34                    |
> | **SG2Loc (Ours)**         | 0.12           | **2.55**       | 0.36                    |
> | **SG2Loc w/ SLAM poses**  | 0.16           | 2.63           | 0.36                    |
>
>
>
> We have added that experiment to the supplement chapter F.
>
> **Q1: How robust is SG2Loc to semantic segmentation errors or to dynamic scenes where object layouts change?**
> 3RScan is designed for localization under strong scene changes. SG2Loc is evaluated in a cross-temporal setting, matching queries to maps captured at different times. As shown in Tables 1-2, it remains robust despite object rearrangements and imperfect semantics. We clarify this in Chapter 5, lines 353–356.
>
> **Q4: How sensitive is the performance to sequence length and particle count?**
> We use ~2500 particles on 3RScan and ~3200 on ScanNet. Raycasting is fully parallelized on the GPU, keeping runtime manageable even with larger particle sets. As shown in Tables 1-2, performance consistently improves with longer sequences (5/10/25/50). We clarify these settings in Chapter 5, lines 411-413.

---

### Comment · Area_Chair_QWmo · 2025-11-28
**Discussions**

Dear Reviewers,

The authors gave their rebuttals to your reviews. Could you please  express your opinions?

Many thanks.

AC

---

### Note · Authors · 2026-02-02

I have read and agree with the venue's withdrawal policy on behalf of myself and my co-authors.

---

### Meta-Review · Area_Chair_S6x4 · 2026-01-02

**Summary:**

The paper initially receives scores of 4, 4, 2, and 2. The AC has read the entire paper, along with all reviews and rebuttals. The reviewers share several important common concerns, particularly regarding limited novelty and insufficient evaluation and comparisons. The AC agrees with the reviewers’ assessments.
Therefore, the AC recommends rejection. The AC also suggests that the authors carefully revise the paper by addressing the reviewers’ comments.

**Reviewer Concerns:**

The reviewers raised several concerns, with the key and common issues summarized below.

1. Reviewer M57c, dsUg, and 2rVs question the novelty of the work, noting the lack of a fundamentally new algorithmic contribution and viewing the approach as an integration of existing techniques. The rebuttal argues that using semantic object identities as the observation signal for pose estimation and integrating them within a sequential particle-filter pipeline is new. The AC agrees that semantic object identities as signals differ from prior approaches, but considers that the core idea and technical components largely build upon existing techniques. The AC suggests that the authors further emphasize the novelty of the motivation and problem formulation.
2. Reviewer M57c and sH4f raise concerns about hyperparameter justification, noting that the method relies on a considerable number of hyperparameters. The rebuttal states that the hyperparameters are unified across benchmarks. The AC believes that even with shared hyperparameters, explicit discussion and ablation studies are still necessary, especially given the method’s dependence on several hyperparameters.
3. Reviewer M57c, dsUg point out missing or unclear implementation details. The rebuttal provides these details, which is expected to address these concerns.
4. Reviewers dsUg and 2rVs note that the system estimates only 4 degrees of freedom, whereas other sota methods address general 6-DoF localization. The rebuttal explains that in sequential localization, gravity direction is typically available from mobile devices, making 4-DoF a practical and commonly used setting. The AC finds this justification reasonable, but notes that supporting 6-DoF localization would further strengthen the generality and impact of the method.
5. Reviewer dsUg, sH4f and 2rVs raise concerns about (i) missing comparisons with certain sota methods, (ii) computational efficiency being lower than retrieval-based approaches, and (iii) limited discussion of other memory-efficient methods. The rebuttal explains that in the keyframing experiments, the method remains real-time in practice when updates are performed at a reduced frame rate. Regarding memory efficiency, the paper includes ACE baselines. However, the AC notes that comparisons with additional sota baselines mentioned by the reviewers should be included in future revisions.
6. Reviewer dsUg and 2rVs question the scope of the evaluation, noting that outdoor datasets are not considered. The rebuttal clarifies that the work focuses on indoor localization, where GPS is unavailable, whereas GPS is typically accessible in outdoor environments.

**Reviewer Scores:**

The AC judges that the reviewers will likely maintain their initial scores, as the concerns regarding novelty, comprehensive comparisons with sota methods, and the reliance on numerous hyperparameters are not fully addressed in the rebuttal. In addition, the reviewers express consistently negative ratings.

---

### Decision · Program_Chairs · 2026-01-26

Reject